# Biochemical and Associated Agronomic Traits in *Gossypium hirsutum* L. under High Temperature Stress

Muhammad Mubashar Zafar [1,2,3,†], Yufang Zhang [1,†], Muhammad Awais Farooq [3,4,†], Arfan Ali [5], Hina Firdous [6], Muhammad Haseeb [6], Sajid Fiaz [7], Amir Shakeel [3,*], Abdul Razzaq [2,8,*] and Maozhi Ren [1,2,9,*]

1. Zhengzhou Research Base, State Key Laboratory of Cotton Biology, School of Agricultural Sciences of Zhengzhou University, Zhengzhou 450000, China; m.mubasharzafar@gmail.com (M.M.Z.); 15093385595@163.com (Y.Z.)
2. Institute of Cotton Research, Chinese Academy of Agricultural Sciences, Anyang 455000, China
3. Department of Plant Breeding and Genetics, University of Agriculture Faisalabad, Faisalabad 38040, Pakistan; awaisfarooq724@gmail.com
4. State Key Laboratory of North China Crop Improvement and Regulation, College of Horticulture, Hebei Agricultural University, Baoding 071000, China
5. FB Genetics Four Brothers Group, Lahore 54000, Pakistan; arfan.alicemb@gmail.com
6. Department of Plant Pathology, University of Agriculture Faisalabad, Faisalabad 38000, Pakistan; hinafirdous72@yahoo.com (H.F.); haseebwahla3360@gmail.com (M.H.)
7. Department of Plant Breeding and Genetics, The University of Haripur, Haripur 22620, Pakistan; sfiaz@uoh.edu.pk
8. Institute of Molecular Biology and Biotechnology, The University of Lahore, Lahore 54792, Pakistan
9. Institute of Urban Agriculture, Chinese Academy of Agriculture Science, Beijing 100049, China
* Correspondence: dramirpbg@gmail.com (A.S.); biolformanite@gmail.com (A.R.); renmaozhi01@caas.cn (M.R.)
† These authors contributed equally to this work.

**Abstract:** Heat tolerance is a physiologically and genetically complex trait regulated by multiple genes. To investigate the genetic basis of heat tolerance, eight parents (five lines and three testers) and their fifteen F1 hybrids were evaluated under normal and high-temperature stress conditions for two consecutive years. Data were recorded for plant height, number of bolls, boll weight, seed cotton yield, ginning out turn (GOT%), $H_2O_2$, catalase, peroxidase, super-oxidase dismutase, total soluble proteins, carotenoids, chlorophyll a & b contents, short fiber index, fiber strength, UHML, micronaire value, reflectance, and uniformity index. Line × tester analysis suggested that the contribution of lines was higher than testers. Non-additive gene action was observed for all studied traits. The variances of SCA were greater than GCA variances for all studied traits revealed that these traits were governed by a few largely dominant genes. Fb-Shaheen, Eagle-2 and JSQ White Gold were found good general combiner whereas the cross Fb-Shaheen × JSQ White Gold was a good specific combiner and revealed significant better parent heterosis for most of the traits during two years under normal and high temperature stress conditions. The information obtained could be utilized in a breeding program for the development of new synthetic varieties of heat tolerance.

**Keywords:** crop stress physiology; heat stress; line x tester; combining ability; heterosis

## 1. Introduction

Cotton (*Gossypium hirsutum* L.) recognized as "White Gold" is a main source of fiber, oil and feed for livestock [1]. The cotton production is adversely affected by both biotic and abiotic stresses. Among abiotic stresses, heat stress is major abiotic stress that lower the cotton growth and production alarmingly [2,3]. In Pakistan, during the period of cotton early growth, the temperature ranges (40–47 °C) in May-June which is very high as compared to other world cotton producing countries where traditional varieties are being grown [4]. The average temperature of cotton growing areas of Pakistan is 37/25 °C. High temperature during flowering periods results in flower shedding, reduced boll weight and

ultimately lower the cotton yield [5]. In Pakistan, average boll weight is 2–3 g which is lower than other cotton producing countries [4]. The boll weight is reduced, when the temperature rises above than 25.5 to 29.5 °C [6]. The coincidence of high temperature with reproductive stages is a major barrier to achieve yield potential in Pakistan. Since the temperature rises up to 47 °C in May-June while accompanying high humidity in July-August which badly effects the cotton reproductive stages [7]. High temperature stress decreases the chlorophyll contents that ultimately reduces the photosynthetic rate as well as translocation of assimilates to reproductive organs and increase senescence [8]. High temperature stress also distorts roots development, stomatal movement, and results in the poor gaseous exchange. The optimum temperature for cotton seed germination (12 °C), root development (30 °C), seedling development and stomatal conductance (28–30 °C), for boll development is (25.5 to 29.5 °C) [6,9]. Every 1 °C rise of temperature in field reduces the seed cotton yield by 110 kg ha$^{-1}$ [10]. Due to the elevated temperature stress, reduction in yields has become a challenging issue particularly for cotton crop grown in arid, semi-arid areas of the world [11]. The detailed information regarding agronomic, ionic, and physiological traits under heat stress is necessary for successful breeding program for yield improvement. Moreover, the knowledge about the inheritance pattern of heat tolerant traits can help breeder for selection of genotype in breeding program [12]. The genetic components governing heat tolerant traits, superior parents and crosses can be assessed by line × tester mating design [3]. The objectives of this research were: (1) to investigate the combining abilities among various morphological, ionic, and physiological characters under normal and heat stress using Line × Tester analysis; (2) based on the identification of the genotypes with best general combining ability (GCA) and specific combining ability (SCA) estimates, to determine whether hybrid breeding may leverage heat tolerance variation in diverse cotton genotypes; and (3) to study the nature of gene action.

## 2. Materials and Methods

### 2.1. Experimental Design

During November 2017, an experiment was conducted to screen 50 cotton genotypes under field condition of Four Brother (FB) Genetics Four Brothers Group, Pakistan and from screening experiment, eight cotton genotypes were selected based on seed cotton yield (SCY) under high temperature conditions [13]. In the next season, they were crossed in L × T mating design. Five heat tolerant genotypes were taken as lines (Ghuari-1, Badar-1, Eagle-2, CCRI-24 and Fb-Shaheen) and three heat sensitive as testers (Fb-Falcon, Fb-Smart 1 and JSQ White Gold). Eight parents and their fifteen F1 hybrids were sown at field research area of FB Genetics Four Brothers Group, Pakistan under two conditions i.e., normal and high temperature stress (5–6 °C above normal for 12 days at 50% flowering) for two consecutive years 2018 and 2019 (Supplementary Table S1). The experiment was conducted in randomized complete block design (RCBD) with three replications following split plot arrangement. The plant × plant and row × row distance was kept at 30 cm and 75 cm, respectively, with a 6 m row length for each genotype under each replication. The seed of selected genotypes was manually sown (dibble method) on furrows in June. The crop was harvested in October each year. The R × R and P × P distance was 75 cm and 30 cm, respectively. The whole experiment was repeated for two consecutive years during normal cotton growing season following the recommended agronomic practices.

### 2.2. Imposition of High Temperature Stress

At 50% flowering, high temperature stress was applied for 12 consecutive days. High temperature stress was applied in September for both consecutive years for 12 days at 50% flowering (Supplementary Table S1). The considerable temperature was raised by constructing tunnel using polythene sheets with help of bamboo arcs and plastic ropes in daytime and uncovered at night. Temperature in the tunnel was measured by using digital thermometer. After the application of high temperature stress, data were recorded

regarding biochemical traits for analysis. Maximum and minimum temperature recorded during 12 days of stress.

### 2.3. Biochemical and Yield-Related Parameters

After 12 days of high temperature exposure, leaf samples were collected from each genotype for the quantification of hydrogen peroxide, catalase, peroxidase, total soluble proteins, chlorophyll contents and carotenoids in the leaves. At maturity, data was collected regarding plant height (PH), number of bolls (NB), boll weight (BW), seed cotton yield (SCY) and GOT% from five plant of each genotype in each replication.

### 2.4. Fiber Quality Traits

A representative sample from seed cotton was weighed and ginning was carried out by single roller ginning machine (Testex, Model: TB510C). The seeds were separated from the lint and ginning out turn was obtained by dividing the weight of lint in a sample by seed cotton weight of the sample, which was expressed in percentage. Lint was further processed to take out the parameters of lint mass per boll, fiber fineness, fiber strength, fiber length, Reflectance, upper half mean length and uniformity ratio with high volume instrument (HVI-900, USTER, Knoxville, TN, USA).

$$GOT\% = \frac{Weight\ of\ lint\ in\ a\ sample}{Weight\ of\ seed\ cotton\ in\ a\ sample} \times 100 \tag{1}$$

### 2.5. Biochemical Traits

Young leaves were collected from the top of the plant for biochemical analysis from each genotype. Enzyme extraction was performed with 0.5 g of cotton leaf sample was cut off with the help of leaf pincher and crushed in chilled condition with 1–2 mL of potassium phosphate buffer (pH 7.8). For grinding of sample mortar and pestle were used with buffer solution. Prepared mixture was then centrifuged for 5 min at 1400 rpm. Residues were discarded and the supernatant was used for biochemical attributes determination by using UV spectrometer at different wavelengths.

### 2.6. Hydrogen Peroxide (µmol/g-FW)

$H_2O_2$ was determine by following protocol purposed by [14]. 0.5 g of cotton leaf was homogenized with 5 mL trichloroacetic acid (TCA) of 0.1% ($W/V$) in pestle and mortar and then centrifuge the homogenate at $14,000 \times g$ for 10 min. Potassium phosphate buffer (0.5 mL; pH 7) and 1ml potassium iodide (KI, 1M) were mixed in 0.5 mL of enzyme extract. Absorbance was taken through a spectrophotometer at 390 nm wavelength (NanoDrop™ 8000 Thermo Fisher Scientific) using deionized water as blank.

### 2.7. Catalase (U/mg Protein)

Enzyme extract (0.1 mL) was mixed with 3ml of reaction mixture, containing 5.9 mM $H_2O_2$ and 50 mM potassium phosphate buffer (7.0 pH). CAT activity was measured at the wavelength of 240 nm [15].

### 2.8. Peroxidase (U/mg Protein)

POD solution containing 0.1 mL enzyme extract, 40 mM hydrogen peroxide ($H_2O_2$), 50 mM potassium phosphate buffer (pH 5) and 20 mM guaiacol. Absorbance was taken through a spectrophotometer at 470 nm wavelength [15].

### 2.9. Total Soluble Proteins (mg/g-FW)

100 µL of enzyme extracted and 5 mL of Bradford reagent mixed. Absorbance was measured at the wavelength of 595 nm by a spectrophotometer [16].

### 2.10. Chlorophyll Contents and Carotenoids Assay

0.5 g of cotton leaf sample was crushed in 8–10 mL of 80% acetone (*v/v*) and then homogenization was carried out through filter paper and absorbance of the final solution was taken at 645 and 663 nm [17]. The chlorophyll a, chlorophyll b and carotenoids were quantified as under.

$$\text{Chlorophyll a}\left(\frac{\mu g}{g}\text{FW}\right) = [12.7\,(\text{OD }663) - 2.69\,(\text{OD }645)] \times \frac{v}{1000\times w}$$

$$\text{Chlorophyll b}\left(\frac{\mu g}{g}\text{FW}\right) = [22.9\,(\text{OD }665) - 4.48\,(\text{OD }663)] \times \frac{V}{1000\times w}$$

where W = weight of leaf sample, V = volume of sample, Em = 2500

$$\text{Carotenoids (mg/g FW)} = \text{Acar}/\text{Em} \times 100$$

$$\text{Acar} = \text{O.D }480 + 0.114\,(\text{O.D }663) - 0.638\,(\text{O.D }645)$$

### 2.11. Statistical Analysis

Collected data were analyzed through analysis of variance as specified by [18] to estimate genetic variability present in parents and crosses. Line × tester analysis of [19] was used for the estimation of combining ability effects of parental genotypes and crosses using R package (Agricolae) in R studio [20] and heterosis effects were calculated by method of [21]. Better parent heterosis was calculated as a percent increase and decrease of hybrid over the better parent.

$$\text{Better parent heterosis (HB)} = (F_1 - BP)/BP, BP = \text{better parent}$$

## 3. Results

### 3.1. Mean Square Line × Tester Analysis for All Traits

During both years, mean square of the line × tester analysis shown significant ($p \leq 0.05$) genetic differences among the genotypes for plant height, total number of bolls, boll weight, seed cotton yield, ginning out turn, $H_2O_2$, MIC value, reflectance, short fiber index, fiber strength, upper half mean length and uniformity index whilst highly significant differences were observed ($p \leq 0.01$) for catalase, peroxidase, superoxide dismutase, total soluble proteins, chlorophyll a & b and carotenoids under normal and heat stress conditions (Tables 1–3). Line × tester analysis indicated that parents were statistically significant ($p \leq 0.05$) for all studied traits while highly significant differences ($p \leq 0.01$) were observed for catalase, peroxidase, superoxide dismutase, total soluble proteins, and chlorophyll a & b under both conditions. Interestingly for parents, during first year, the reflectance showed non-significant results under normal conditions, whilst uniformity index revealed non-significant results under heat stress conditions during both years. Crosses were also found significant for all of the investigated traits. The analysis of variance for heat stress × year showed non-significant interaction for all traits except $H_2O_2$, peroxidase, chlorophyll a & b contents, fiber strength and micronaire value. The genotypes × year interaction showed non-significant interaction for all traits except SOD, TSP, chlorophyll a content, fiber strength, micronaire value and uniformity index. The heat stress × year × genotypes interaction showed non-significant interaction for all traits except $H_2O_2$, SOD, TSP, chlorophyll b contents, short fiber index, fiber strength micronaire value and uniformity index (Tables 4–7).

**Table 1.** List of lines, testers and their cross combinations used in experiment.

| Lines | Testers | Crosses | Crosses | Crosses |
|---|---|---|---|---|
| Ghuari-1 | Fb-Falcon | Ghuari-1 × Fb-Falcon | Badar-1 × JSQ White Gold | CCRI-24 × Fb-Smart1 |
| Badar-1 | Fb-Smart 1 | Ghuari-1 × Fb-Smart1 | Eagle-2 × Fb-Falcon | CCRI-24 × JSQ White Gold |
| Eagle-2 | JSQ White Gold | Ghuari-1 × JSQ White Gold | Eagle-2 × Fb-Smart1 | Fb-Shaheen × Fb-Falcon |
| CCRI-24 | | Badar-1 × Fb-Falcon | Eagle-2 × JSQ White Gold | Fb-Shaheen × Fb-Smart1 |
| Fb-Shaheen | | Badar-1 × Fb-Smart1 | CCRI-24 × Fb-Falcon | Fb-Shaheen × JSQ White Gold |

**Table 2.** Mean squares value from ANOVA of Heat × Year × Genotypes interaction.

| S.O. V | PH | TNB | BW | SCY | GOT% | $H_2O_2$ | CAT | POD | SOD |
|---|---|---|---|---|---|---|---|---|---|
| Replication. (R) | 32.56 | 38.07 | 0.20 | 127.56 | 128.557 | 9.541 | 708 | 99.9 | 7.9 |
| Treatment. (T) | 1103.48 | 1715.93 * | 0.13 | 4121.16 * | 128.5 * | 915.8 * | 1432 ** | 1870 ** | 7934 ** |
| Error. R × T | 16.68 | 5.74 | 0.03 | 32.39 | 0.976 | 0.84 | 147 | 6.88 | 6.7 |
| Geno. (G) | 287.29 ** | 72.95 ** | 0.38 ** | 481.6 ** | 114.8 ** | 52.2 ** | 4379 ** | 979.8 ** | 261.1 ** |
| T × G | 56.46 ** | 23.36 ** | 0.09 ** | 109.7 ** | 30.0 * | 9.1 * | 1150 ** | 639.6 ** | 186.4 ** |
| Error R ×T× G | 17.92 | 7.46 | 0.04 | 14.99 | 14.029 | 4.2 | 53 | 18.3 | 35.4 |
| Year (Y) | 8.27 * | 8.91 * | 0.06 | 28.96 | 5.223 | 17.1 ** | 57 ** | 0.021 | 4.0 |
| T × Y | 3.96 | 0.23 | 0.05 | 16.44 | 1.962 | 22.1 ** | 2 | 54.3 ** | 15.3 |
| G × Y | 2.18 | 1.61 | 0.01 | 5.44 | 1.552 | 1.278 | 5 | 9.45 * | 15.6 * |
| T × G × Y | 1.37 | 1.26 | 0.03 | 7.33 | 1.587 | 1.8 * | 4 | 5.32 | 21.0 ** |
| Error. R × T × G × Y | 2.16 | 1.34 | 0.02 | 9.05 | 1.527 | 0.980 | 6 | 4.77 | 9.0 |

*, significance (α = 0.05); **, highly significant (α = 0.01); PH, plant height (cm); TNB, number of bolls; BW, boll weight (g); SCY, seed cotton yield (g); GOT, ginning outturn (%); $H_2O_2$, hydrogen peroxide (μmol/g); CAT, catalase (U mg$^{-1}$ protein); POD, peroxidase (U mg$^{-1}$ protein); SOD, super-oxidase dismutase (U mg$^{-1}$ protein).

**Table 3.** Mean squares value from ANOVA of Heat × Year × Genotypes interaction.

| S.O. V | TSP | Chloro. Con | | Caro. | MIC | RD | SF | STR | UHML | UI |
|---|---|---|---|---|---|---|---|---|---|---|
| | | a | B | | | | | | | |
| Replication. (R) | 27.01 | 0.03 | 0.003 | 0.00 | 0.15 | 23.2 | 0.03 | 0.12 | 0.34 | 1.571 |
| Treatment. (T) | 5988 * | 15.1 * | 1.020 ** | 8.26 * | 0.082 | 411 * | 2.75 | 6.5 | 54.3 | 152.2 * |
| Error. R × T | 32.98 | 7.04 | 2.17 | 0.00 | 0.000 | 0.33 | 0.27 | 0.10 | 2.1 | 0.20 |
| Geno. (G) | 40.16 ** | 0.19 ** | 0.02 ** | 0.02 ** | 2.3 ** | 22.0 ** | 1.73 ** | 45.2 ** | 18.4 ** | 18.1 ** |
| T × G | 28.60 ** | 0.10 ** | 0.009 ** | 0.009 ** | 0.36 ** | 18.6 ** | 2.907 ** | 21.7 ** | 9.52 ** | 19.8 ** |
| Error R × T × G | 2.35 | 0.00 | 0.001 | 0.001 | 0.090 | 4.720 | 0.293 | 3.740 | 1.73 | 3.370 |
| Year (Y) | 16.26 ** | 1.35 | 0.016 ** | 0.004 ** | 0.12 | 5.918 | 0.374 ** | 105.6 ** | 1.06 | 1.962 |
| T × Y | 0.04 | 0.05 ** | 0.006 ** | 0.001 | 1.02 ** | 0.918 | 0.074 | 26.176 ** | 2.63 | 3.962 |
| G × Y | 2.41 ** | 0.00 | 5.01 | 0.000 | 0.10 * | 2.759 | 0.02 | 8.26 ** | 2.54 | 10.9 ** |
| T × G × Y | 2.47 ** | 0.001 | 7.94 * | 0.000 | 0.10 * | 2.214 | 0.079 * | 11.7 ** | 2.92 | 11.0 ** |
| Error. R × T × G × Y | 1.06 | 0.001 | 3.53 | 0.000 | 0.05 | 2.397 | 0.042 | 3.584 | 2.12 | 3.10 |

*, significance (α = 0.05); **, highly significant (α = 0.01); TSP, total soluble protein (mg g$^{-1}$ FW); Chl a & b, chlorophyll contents a & b (mg g$^{-1}$ FW); Caro., carotenoids (mg g$^{-1}$ FW); SF, short fiber contents (%); STR, fiber strength (g/tex); MIC, MIC value (unit); RD, reflectance; UI, uniformity index (%); UHML, upper half mean length (mm).

**Table 4.** First year mean squares of line × tester analysis for various characters under normal and high temperature stress conditions.

| S.O. V | Trt | PH | TNB | BW | SCY | GOT% | H$_2$O$_2$ | CAT | POD | SOD |
|---|---|---|---|---|---|---|---|---|---|---|
| Replication | N | 0.7826 | 8.95 | 0.0020 | 10.52 | 37.62 | 1.42 | 65.04 | 79.04 | 4.14 |
| | HT | 25.42 | 40.76 | 0.25 | 176.47 | 32.72 | 1.61 | 604.83 | 35.30 | 5.22 |
| Genotype | N | 78.44 * | 35.9 * | 0.158 * | 136.78 * | 47.0 * | 17.74 * | 1606.8 ** | 779.5 ** | 159.6 * |
| | HT | 87.02 * | 14.83 * | 0.15 * | 169.93 * | 31.65 * | 19.49 * | 1252.30 ** | 83.83 ** | 106.03 ** |
| Crosses | N | 73.7 * | 35.1 * | 0.135 * | 105.63 * | 27.6 * | 18.1 * | 1664.4 ** | 833.0 ** | 140.8 * |
| | HT | 49.91 * | 11.03 * | 0.083 * | 160.32 * | 31.63 * | 23.36 * | 1489.82 ** | 82.71 ** | 113.85 ** |
| Line | N | 69.3 * | 29.1 * | 0.148 * | 153.7 ** | 26.3 * | 9.0 * | 2210.1 ** | 1327 ** | 202.5 * |
| | HT | 52.15 * | 11.55 * | 0.10 * | 176.45 * | 57.56 * | 20.19 * | 611.73 ** | 54.78 ** | 117.86 * |
| Tester | N | 96.07 * | 24.5 * | 0.181 * | 82.09 | 31.7 * | 4.3 * | 296.7 ** | 593.6 ** | 112.3 * |
| | HT | 33.81 * | 6.41 * | 0.07 * | 155.01 * | 34.87 * | 25.03 * | 2594.39 * | 76.22 * | 95.90 * |
| L × T | N | 70.3 * | 40.7 * | 0.117 * | 87.4 * | 27.3 * | 21.1 * | 1733.58 ** | 645.7 ** | 117.2 * |
| | HT | 52.81 * | 11.92 * | 0.08 * | 153.59 * | 17.86 * | 24.53 * | 1652.73 ** | 98.29 ** | 86.34 * |
| Parent | N | 62.4 * | 39.4 * | 0.141 * | 210.9 ** | 28.4 * | 17.6 * | 1684.08 ** | 735.3 * | 173.9 * |
| | HT | 166.2 * | 17.64 * | 0.13 * | 212.2 * | 29.49 * | 13.07 * | 921.10 ** | 93.44 * | 97.53 * |
| CrovsPar | N | 256.7 ** | 22.5 * | 0.594 * | 54.04 * | 120.2 * | 13.0 * | 260.13 ** | 339.3 ** | 321.4 * |
| | HT | 52.14 * | 48.45 * | 1.16 * | 8.16 * | 81.95 * | 10.27 * | 245.82 ** | 32.23 ** | 56.08 * |
| Error | N | 13.9 | 5.19 | 0.0365 | 11.65 | 8.92 | 2.9 | 22.25 | 22.47 | 43.94 |
| | HT | 8.64 | 3.64 | 0.02 | 17.38 | 8.85 | 2.27 | 52.33 | 6.81 | 20.47 |
| Total | N | 2033.4 | 913.1 | 4.287 | 3276.3 | 1267.9 | 456.8 | 35,905.9 | 17,723.3 | 4482.7 |
| | HT | 2130.27 | 449.05 | 4.05 | 4297.49 | 293.74 | 480.59 | 21,961.7 | 2029.4 | 2788.36 |

*, significance (α = 0.05); **, highly significant (α = 0.01); PH, plant height (cm); TNB, number of bolls; BW, boll weight (g); SCY, seed cotton yield (g); GOT, ginning outturn (%); H$_2$O$_2$, hydrogen peroxide (μmol/g); CAT, catalase (U mg$^{-1}$ protein); POD, peroxidase (U mg$^{-1}$ protein); SOD, super-oxidase dismutase (U mg$^{-1}$ protein).

*3.2. Genetic Components and Proportional Contributions of Lines, Testers and Their Interaction to Total Variance under Normal and Heat Stress Conditions*

During both years, the variance due to SCA was higher and more significant than GCA for all traits reflecting the dominant role of non-additive type of gene action under both conditions (Tables 8 and 9). The decrease in non-additive gene action value were observed for PH, BW, TNB, SCY, RD, UI, CAT, POD, SOD, chlorophyll contents and carotenoids under heat stress whilst GOT%, STR, MIC, SF, UHML, H$_2$O$_2$, and TSP revealed higher value for non-additive gene action under heat stress conditions (Tables 8 and 9). During both years, the contribution of line × tester interaction was higher for most of the agronomic (PH, TNB, BW and GOT), biochemical (H$_2$O$_2$, chlorophyll a & b, peroxidase, superoxide dismutase, total soluble protein and carotenoids), and fiber quality traits (STR, MIC, SF and UHML) under heat stress conditions whilst the line × tester interaction was lower as compared to lines or testers for seed cotton yield, reflectance and uniformity index (Tables 8 and 9). During both years, the lines (females) showed higher contribution than testers for most of the studied such as likse PH, TNB, BW, GOT, SCY, STR, MIC, UHML, UI, POD, SOD, TSP, and chlorophyll contents under both conditions (Tables 8 and 9).

**Table 5.** First year mean squares of line × tester analysis for various characters under normal and high temperature stress conditions.

| S.O. V | Trt | TSP | Chloro. Con | | Caro. | MIC | RD | SF | STR | UHML | UI |
|---|---|---|---|---|---|---|---|---|---|---|---|
| | | | a | b | | | | | | | |
| Replication | N | 0.208 | 0.005 | 0.0030 | 0.0063 | 0.0176 | 3.61 | 0.182 | 0.003 | 0.087 | 0.70 |
| | HT | 38.53 | 0.02 | 0.0002 | 0.002 | 0.02 | 6.51 | 2.63 | 1.22 | 0.005 | 3.84 |
| Genotype | N | 3.10 ** | 0.12 ** | 0.013 ** | 0.0075 ** | 1.009 * | 11.08 * | 1.87 * | 18.6 * | 6.133 * | 16.09 * |
| | HT | 35.48 ** | 0.03 ** | 0.003 ** | 0.01 ** | 0.60 ** | 8.45 * | 8.26 * | 17.04 * | 0.55 * | 5.15 * |
| Crosses | N | 3.36 ** | 0.129 ** | 0.0123 ** | 0.0073 ** | 0.752 * | 13.3 * | 2.08 * | 20.75 * | 3.59 * | 14.84 * |
| | HT | 39.33 ** | 0.02 ** | 0.003 ** | 0.01 ** | 0.52 ** | 8.42 * | 5.86 * | 11.88 * | 0.49 * | 4.14 * |
| Line | N | 3.60 ** | 0.088 ** | 0.008 * | 0.0041 * | 1.892 ** | 8.95 | 3.80 * | 51.23 ** | 6.36 * | 34.7 ** |
| | HT | 20.91 ** | 0.01 ** | 0.001 ** | 0.004 ** | 0.678 * | 11.12 * | 3.00 * | 12.02 * | 0.26 * | 5.08 * |
| Tester | N | 3.91 ** | 0.262 ** | 0.014 ** | 0.0146 ** | 0.364 * | 6.39 * | 3.75 * | 14.9 * | 1.16 | 8.34 * |
| | HT | 10.54 ** | 0.02 ** | 0.003 ** | 0.02 ** | 0.23 ** | 15.66 * | 1.31 * | 16.86 * | 0.67 * | 4.40 * |
| L× T | N | 3.11 ** | 0.116 ** | 0.013 ** | 0.0071 ** | 0.279 * | 17.24 * | 0.79 * | 6.96 * | 2.81 * | 6.52 * |
| | HT | 55.74 ** | 0.02 ** | 0.003 ** | 0.01 ** | 0.51 * | 5.25 * | 8.42 * | 10.56 * | 0.57 * | 3.60 * |
| Parent | N | 2.84 ** | 0.117 ** | 0.0185 ** | 0.0081 ** | 0.29 * | 4.41 | 0.52 | 12.3 * | 6.96 * | 11.5 * |
| | HT | 30.67 ** | 0.04 ** | 0.004 ** | 0.01 ** | 0.41 ** | 8.41 * | 13.56 | 4.01 * | 0.16 * | 4.72 |
| CrovsPar | N | 1.14 * | 0.073 ** | 0.0045 ** | 0.0061 ** | 9.60 ** | 26.35 * | 8.49 * | 32.72 ** | 35.89 ** | 65.50 * |
| | HT | 15.16 * | 0.06 ** | 0.007 ** | 0.017 ** | 3.17 ** | 9.15 * | 4.88 * | 180.51 ** | 3.94 * | 22.28 * |
| Error | N | 0.195 | 0.0075 | 0.0010 | 0.0005 | 0.103 | 5.63 | 0.319 | 2.05 | 0.95 | 1.88 |
| | HT | 2.56 | 0.003 | 0.001 | 0.002 | 0.06 | 1.34 | 1.04 | 2.91 | 0.05 | 1.06 |
| Total | N | 72.73 | 2.87 | 0.3314 | 0.1836 | 24.49 | 371.3 | 48.53 | 455.12 | 156.01 | 396.25 |
| | HT | 875.40 | 0.67 | 0.10 | 0.30 | 14.73 | 221.87 | 207.27 | 440.23 | 13.26 | 140.51 |

*, significance (α = 0.05); **, highly significant (α = 0.01); TSP, total soluble protein (mg g$^{-1}$ FW); Chl a & b, chlorophyll contents a & b (mg g$^{-1}$ FW); Caro., carotenoids (mg g$^{-1}$ FW); SF, short fiber contents (%); STR, fiber strength (g/tex); MIC, MIC value (unit); RD, reflectance; UI, uniformity index (%); UHML, upper half mean length (mm).

### 3.3. General and Specific Combining Ability Effects under Normal and High Temperature Stress Conditions

3.3.1. GCA and SCA Effects for Yield Contributing Traits

During both years, positive and significant GCA effects for PH, BW and SCY were observed for FB-SHAHEEN under both conditions. Among testers, JSQ WHITE GOLD revealed positive and significant GCA estimates for SCY under both conditions for consecutive years. The parent Eagle-2 and CCRI-24 revealed positively significant GCA effects for number of bolls per plant and GOT%, respectively, under both conditions. During both years, hybrid Eagle-2 × FB-Falcon and FB-Shaheen × JSQ White Gold showed highly significant and positive SCA estimates for PH, SCY, BW under both conditions. For GOT%, the significant and positive SCA effects were observed in Ghuari-1 × FB-Falcon under control conditions whilst under heat stress FB-Shaheen × FB-Falcon and CCRI-24 × JSQ White Gold showed positive and significant SCA effects (Tables S2 and S3).

**Table 6.** Second year mean squares of line × tester analysis for various characters under normal and high temperature stress conditions.

| S.O. V | Trt | PH | TNB | BW | SCY | GOT% | $H_2O_2$ | CAT | POD | SOD |
|---|---|---|---|---|---|---|---|---|---|---|
| Replication | N | 0.54 | 0.610 | 0.0557 | 5.5652 | 17.78 | 0.947 | 41.515 | 13.915 | 11.30 |
| | HT | 22.54 | 4.76 | 0.02 | 13.69 | 43.63 | 7.52 | 199.51 | 1.84 | 4.57 |
| Genotype | N | 85.74 * | 36.13 * | 0.120 * | 129.05 ** | 39.29 * | 15.68 * | 1453.0 ** | 690.60 ** | 130.40 * |
| | HT | 96.07 * | 12.29 * | 0.11 * | 168.39 * | 30.15 * | 11.58 * | 1226.18 ** | 79.05 ** | 88.41 * |
| Crosses | N | 84.70 * | 33.78 * | 0.094 * | 99.88 ** | 27.25 * | 15.86 * | 1498.7 ** | 748.61 ** | 101.8 * |
| | HT | 50.56 * | 8.75 * | 0.14 * | 163.83 ** | 33.67 * | 12.32 * | 1427.8 ** | 103.73 ** | 69.19 * |
| Line | N | 93.5 * | 24.25 * | 0.116 * | 214.55 ** | 21.32 * | 8.142 * | 1872.5 ** | 1254.0 ** | 105.98 * |
| | HT | 65.55 * | 12.25 * | 0.05 * | 159.79 ** | 67.19 * | 16.56 * | 562.07 ** | 71.37 ** | 134.36 * |
| Tester | N | 84.7 * | 37.64 * | 0.110 * | 27.741 * | 32.83 * | 20.52 * | 380.86 ** | 585.99 ** | 198.18 ** |
| | HT | 30.13 * | 7.66 * | 0.03 * | 173.77 * | 36.53 * | 15.89 * | 2481.4 ** | 105.82 ** | 15.89 ** |
| L × T | N | 80.2 * | 37.58 * | 0.079 * | 60.587 * | 28.83 * | 18.55 * | 1591.3 ** | 536.56 ** | 75.69 * |
| | HT | 48.17 * | 7.27 * | 0.21 * | 163.37 * | 16.18 * | 9.31 * | 1597.2 ** | 119.39 ** | 9.31 * |
| Parent | N | 74.4 * | 39.83 * | 0.113 * | 196.5 ** | 19.72 * | 14.34 * | 1520.1 ** | 629.92 ** | 149.37 ** |
| | HT | 190.67 * | 9.52 * | 0.05 * | 197.19 * | 19.33 * | 9.70 * | 947.47 ** | 34.18 ** | 9.70 ** |
| CrovsPar | N | 179.4 ** | 43.28 * | 0.524 ** | 64.826 * | 344.75 | 22.48 * | 343.1 ** | 303.15 ** | 397.34 ** |
| | HT | 71.10 ** | 81.49 * | 0.19 * | 30.71 * | 56.73 * | 14.31 * | 354.36 ** | 47.49 ** | 14.31 * |
| Error | N | 10.43 | 5.29 | 0.0270 | 7.084 | 7.044 | 2.46 | 10.29 | 9.6432 | 10.050 |
| | HT | 7.32 | 2.99 | 0.03 | 10.66 | 6.33 | 2.80 | 29.78 | 7.19 | 15.09 |
| Total | N | 2116.70 | 912.09 | 3.2965 | 3000.55 | 1037.2 | 400.12 | 32,234.5 | 15,419.33 | 3101.25 |
| | HT | 2297.15 | 341.31 | 3.25 | 3953.95 | 846.38 | 323.94 | 27,830.7 | 1899.09 | 2281.62 |

*, significance (α = 0.05); **, highly significant (α = 0.01); PH, plant height (cm); TNB, number of bolls; BW, boll weight (g); SCY, seed cotton yield (g); GOT, ginning outturn (%); $H_2O_2$, hydrogen peroxide (μmol/g); CAT, catalase (U mg$^{-1}$ protein); POD, peroxidase (U mg$^{-1}$ protein); SOD, super-oxidase dismutase (U mg$^{-1}$ protein).

### 3.3.2. GCA and SCA for Antioxidants

The parents, FB-SHAHEEN and JSQ WHITE GOLD showed positive and significant GCA estimates for $H_2O_2$, CAT, POD under studied conditions. During both years, significant and positive GCA estimates for SOD, TSP, chlorophyll contents and carotenoids were observed in FB-SHAHEEN under heat stress conditions. Eagle-2 × Fb-Smart1 and FB-Shaheen × JSQ White Gold showed positive and significant SCA estimates for $H_2O_2$ under both conditions. Based on the results of SCA effects for hybrids, FB-SHAHEEN × JSQ WHITE GOLD showed highest positive significant SCA value for POD, SOD and TSP under both conditions (Tables S2–S5).

### 3.3.3. GCA and SCA for Fiber Quality Traits

For MIC, UI and UHML good general combiner was FB-SHAHEEN under normal and heat stress conditions during both years. Among all lines and testers, Eagle-2 and JSQ WHITE GOLD were good general combiner for SF under heat stress conditions. During both years, the cross combination of Badar-1 × Fb-Falcon and Eagle-2 × JSQ White Gold exhibited positive and significant SCA effects for MIC, SF and UI under both conditions. During both years, the cross combinations of Ghuari-1 × FB-Smart1 and FB-Shaheen × FB-Falcon revealed positive and significant SCA effects for fiber strength under heat stress conditions. During the two years, Eagle-2 × JSQ White Gold and CCRI-24 × Fb-Smart1 had positive and significant SCA estimates for UHML under heat stress conditions. During 1st year, all crosses showed non-significant or negative SCA effects for uniformity index whilst for 2nd year, the Eagle-2 × JSQ White Gold and Eagle-2 × JSQ White Gold had significant and positive SCA effects under normal conditions (Tables S4 and S5).

**Table 7.** Second year mean squares of line × tester analysis for various characters under normal and high temperature stress conditions.

| S.O. V | Trt | TSP | Chloro. Con | | Caro. | MIC | RD | SF | STR | UHML | UI |
|---|---|---|---|---|---|---|---|---|---|---|---|
| | | | a | b | | | | | | | |
| Replication | N | 13.91 | 0.008 | 0.0000 | 0.0015 | 0.062 | 5.495 | 0.0078 | 0.0035 | 5.56 | 0.002 |
| | HT | 22.40 | 0.003 | 0.001 | 0.001 | 0.07 | 8.09 | 6.28 | 3.18 | 0.06 | 0.06 |
| Genotype | N | 690.6 ** | 0.126 ** | 0.011 ** | 0.0072 * | 0.82 * | 19.74 * | 38.32 * | 18.63 * | 11.29 * | 28.33 * |
| | HT | 32.25 ** | 0.03 ** | 0.004 ** | 0.008 * | 0.44 * | 6.44 * | 7.75 * | 12.93 * | 0.64 * | 10.37 * |
| Crosses | N | 748.6 ** | 0.132 ** | 0.008 ** | 0.006 * | 0.634 * | 21.68 * | 25.85 * | 20.75 * | 10.50 * | 32.91 * |
| | HT | 35.09 ** | 0.021 ** | 0.004 ** | 0.009 * | 0.21 * | 6.03 * | 5.42 * | 8.67 * | 0.65 * | 9.66 * |
| Line | N | 1254.0 ** | 0.086 ** | 0.005 * | 0.006 * | 1.23 * | 18.22 * | 56.84 * | 51.23 ** | 16.71 * | 33.62 * |
| | HT | 14.11 ** | 0.015 ** | 0.004 * | 0.007 * | 0.36 * | 5.39 * | 7.16 * | 9.78 * | 0.64 * | 18.34 |
| Tester | N | 585.99 ** | 0.27 ** | 0.014 ** | 0.005 * | 0.369 * | 23.28 * | 16.02 | 14.92 * | 0.208 | 28.35 * |
| | HT | 0.76 ** | 0.02 ** | 0.005 ** | 0.019 * | 0.02 * | 13.84 * | 2.00 * | 4.62 * | 0.79 * | 13.94 * |
| L× T | N | 536.5 ** | 0.119 ** | 0.009 | 0.006 * | 0.400 * | 23.01 * | 12.8 | 6.96 * | 9.96 | 33.69 * |
| | HT | 54.17 ** | 0.02 ** | 0.002 * | 0.008 * | 0.18 * | 4.39 * | 5.40 * | 9.13 * | 0.62 * | 3.95 * |
| Parent | N | 629.9 ** | 0.113 ** | 0.019 ** | 0.0075 * | 0.17 * | 15.35 * | 20.59 * | 12.38 * | 11.364 * | 7.89 * |
| | HT | 25.37 ** | 0.04 ** | 0.005 ** | 0.007 * | 0.29 * | 6.44 * | 8.81 * | 12.35 * | 0.19 * | 3.10 |
| CrovsPar | N | 303.15 ** | 0.1320 ** | 0.0000 | 0.0222 ** | 8.134 ** | 23.270 * | 337.0 ** | 32.72 ** | 21.82 * | 107.32 |
| | HT | 40.75 ** | 0.02 ** | 0.004 * | 0.01 ** | 4.72 ** | 12.12 * | 32.89 * | 76.68 * | 3.60 * | 71.15 * |
| Error | N | 9.643 | 0.0052 | 0.0008 | 0.0007 | 0.072 | 6.10 | 6.94 * | 2.05 | 3.36 | 7.109 |
| | HT | 3.03 | 0.004 | 0.0007 | 0.003 | 0.04 | 1.37 | 2.01 | 2.88 | 0.07 | 3.04 |
| Total | N | 15,419.33 | 2.9067 | 0.2748 | 0.1757 | 19.89 | 574.18 | 995.91 | 455.12 | 327.99 | 779.78 |
| | HT | 798.60 | 0.68 | 0.11 | 0.25 | 10.73 | 179.81 | 220.91 | 351.05 | 15.79 | 295.11 |

*, significance ($\alpha = 0.05$); **, highly significant ($\alpha = 0.01$); TSP, total soluble protein (mg g$^{-1}$ FW); Chl a & b, chlorophyll contents a & b (mg g$^{-1}$ FW); Caro., carotenoids (mg g$^{-1}$ FW); SF, short fiber contents (%); STR, fiber strength (g/tex); MIC, MIC value (unit); RD, reflectance; UI, uniformity index (%); UHML, upper half mean length (mm).

### 3.3.4. Better Parent Heterosis Effects under Normal and Heat Stress Conditions

During both years, for plant height, Eagle-2 × JSQ White Gold, Eagle-2 × Fb-Smart1, FB-Shaheen × FB-Falcon and CCRI-24 × Fb-Smart 1 had significant and negative heterosis effects under both conditions. During both years, Fb-Shaheen × JSQ White Gold exhibited significant and positive heterosis effects for boll weight and SCY under both conditions. The positive and significant heterosis effects for H$_2$O$_2$, were observed for Fb-Shaheen × JSQ White Gold under heat stress conditions. The crosses of Badar-1 × Fb-Falcon, Badar-1 × Fb-Smart1, Badar-1 × JSQ White Gold and Fb-Shaheen × Fb-Smart1 showed positive and significant heterosis effects for short fiber index under heat stress conditions during both years. During both years, for fiber strength and uniformity index the hybrids of Eagle-2 × JSQ White Gold, CCRI-24 × JSQ White Gold, Fb-Shaheen × JSQ White Gold and Fb-Shaheen × Fb-Smart1 revealed significant and positive heterosis estimates under heat stress conditions (Tables 10–13).

**Table 8.** First year genetic components and proportional contribution of lines, testers and their interactions to total variance under normal and high temperature stress conditions.

| Characters | | Trt | $\delta^2$GCA | $\delta^2$SCA | $\delta^2$A | $\delta^2$D | Contribution of Lines | Contribution of Tester | Contribution of L × T |
|---|---|---|---|---|---|---|---|---|---|
| PH | | N | 0.1791 | 28.338 | 0.7165 | 113.35 | 26.86 | 18.62 | 54.52 |
| | | HT | −0.15 | 23.04 | −0.61 | 92.19 | 29.86 | 9.68 | 60.46 |
| TNB | | N | −0.2988 | 16.954 | −1.1953 | 67.81 | 23.70 | 9.98 | 66.32 |
| | | HT | −0.04 | 3.75 | −0.19 | 15.02 | 29.93 | 8.30 | 61.77 |
| BW | | N | 0.000 | 0.038 | 0.003 | 0.1528 | 31.33 | 19.07 | 49.60 |
| | | HT | 0.000 | 0.020 | 0.001 | 0.09 | 35.43 | 12.40 | 52.17 |
| GOT | | N | 0.018 | 8.359 | 0.071 | 33.438 | 27.19 | 16.36 | 56.45 |
| | | HT | 0.36 | 73.06 | 1.43 | 292.24 | 31.44 | 13.81 | 54.74 |
| SCY | | N | 0.9650 | 37.506 | 3.860 | 150.02 | 41.60 | 11.10 | 47.30 |
| | | HT | 0.73 | 5.22 | 2.92 | 20.89 | 51.99 | 15.75 | 32.26 |
| STR | | N | 0.731 | 2.368 | 2.924 | 9.475 | 70.54 | 10.27 | 19.18 |
| | | HT | 0.07 | 3.53 | 0.28 | 14.11 | 28.92 | 20.28 | 50.80 |
| MIC | | N | 0.025 | 0.087 | 0.100 | 0.348 | 71.84 | 6.92 | 21.24 |
| | | HT | 0.000 | 0.22 | 0.001 | 0.88 | 36.93 | 6.50 | 56.57 |
| SF | | N | 0.068 | 0.223 | 0.271 | 0.891 | 52.27 | 25.78 | 21.95 |
| | | HT | −0.004 | 0.26 | −0.015 | 1.04 | 15.04 | 19.19 | 65.77 |
| RD | | N | −0.2078 | 4.8735 | −0.831 | 19.494 | 19.21 | 6.85 | 73.94 |
| | | HT | 0.17 | 1.84 | 0.67 | 7.37 | 37.75 | 26.57 | 35.67 |
| UHML | | N | 0.041 | 0.970 | 0.165 | 3.881 | 50.59 | 4.62 | 44.80 |
| | | HT | −0.13 | 3.65 | −0.54 | 14.64 | 14.65 | 3.19 | 82.16 |
| UI | | N | 0.441 | 2.012 | 1.765 | 8.046 | 66.86 | 8.03 | 25.10 |
| | | HT | 0.03 | 1.29 | 0.11 | 5.19 | 35.10 | 15.18 | 49.7 |
| H$_2$O$_2$ | | N | −0.159 | 8.603 | −0.639 | 34.413 | 14.19 | 19.17 | 66.64 |
| | | HT | −0.06 | 11.15 | −0.25 | 44.62 | 24.69 | 15.30 | 60 |
| CAT | | N | −3.665 | 856.90 | −14.661 | 3427.6 | 37.94 | 2.55 | 59.52 |
| | | HT | −8.64 | 80.89 | −34.56 | 3211.58 | 11.73 | 24.88 | 63.39 |
| POD | | N | 9.935 | 307.55 | 39.743 | 1230.20 | 45.53 | 10.18 | 44.29 |
| | | HT | −0.82 | 45.93 | −3.31 | 183.71 | 18.93 | 13.17 | 67.91 |
| SOD | | N | 1.255 | 40.143 | 5.020 | 160.57 | 41.07 | 11.39 | 47.54 |
| | | HT | 1.46 | 32.40 | 5.83 | 129.60 | 43.63 | 12.04 | 44.33 |
| TSP | | N | 0.013 | 1.465 | 0.053 | 5.862 | 30.56 | 16.60 | 52.84 |
| | | HT | −0.87 | 26.89 | −3.48 | 107.55 | 15.19 | 3.83 | 80.98 |
| Chlo.C | a | N | 0.000 | 0.053 | 0.0027 | 0.212 | 19.52 | 29.07 | 51.41 |
| | b | N | −0.000 | 0.006 | −0.000 | 0.025 | 19.36 | 16.77 | 63.87 |
| | a | HT | −0.000 | 0.009 | −0.000 | 0.038 | 17.88 | 16.40 | 65.71 |
| | b | HT | −0.00 | 0.001 | −0.000 | 0.0056 | 15.79 | 12.28 | 71.93 |
| Caro. | | N | 0.000 | 0.003 | 0.000 | 0.0129 | 16.13 | 28.50 | 55.37 |
| | | HT | −0.000 | 0.004 | −0.000 | 0.02 | 11.64 | 21.12 | 67.24 |

PH, plant height (cm); TNB, number of bolls; BW, boll weight (g); SCY, seed cotton yield (g); GOT, ginning outturn (%); H$_2$O$_2$, hydrogen peroxide (μmol/g); CAT, catalase (U mg$^{-1}$ protein); POD, peroxidase (U mg$^{-1}$ protein); SOD, super-oxidase dismutase (U mg$^{-1}$ protein); TSP, total soluble protein (mg g$^{-1}$ FW); Chl a & b, chlorophyll contents a & b (mg g$^{-1}$ FW); Caro., carotenoids (mg g$^{-1}$ FW); SF, short fiber contents (%); STR, fiber strength (g/tex); MIC, MIC value (unit); RD, reflectance; UI, uniformity index (%); UHML, upper half mean length (mm).

**Table 9.** Second year genetic components and proportional contribution of lines, testers, and their interactions to total variance under normal and high temperature stress conditions.

| Characters | | Trt | $\delta^2$GCA | $\delta^2$SCA | $\delta^2$A | $\delta^2$D | Contribution of Lines | Contribution of Tester | Contribution of L × T |
|---|---|---|---|---|---|---|---|---|---|
| PH | | N | 0.234 | 34.96 | 0.939 | 139.84 | 31.55 | 14.29 | 54.16 |
| | | HT | 0.13 | 20.84 | 0.51 | 83.37 | 37.04 | 8.51 | 54.44 |
| TNB | | N | −0.201 | 15.72 | −0.806 | 62.914 | 20.51 | 15.92 | 63.57 |
| | | HT | 0.08 | 2.02 | 0.32 | 8.08 | 40.01 | 12.5 | 47.47 |
| BW | | N | 0.000 | 0.025 | 0.0031 | 0.100 | 35.08 | 16.66 | 48.26 |
| | | HT | −0.003 | 0.09 | −0.015 | 0.37 | 9.71 | 3.71 | 86.58 |
| GOT | | N | −0.083 | 10.414 | −0.333 | 41.656 | 22.36 | 17.21 | 60.44 |
| | | HT | 0.02 | 77.12 | 0.09 | 308.51 | 27.87 | 15.15 | 56.98 |
| SCY | | N | 2.084 | 26.564 | 8.336 | 106.25 | 61.37 | 3.97 | 34.66 |
| | | HT | 0.92 | 5.66 | 3.71 | 22.65 | 57.03 | 15.50 | 27.47 |
| STR | | N | 0.691 | 3.120 | 2.764 | 12.48 | 62.81 | 8.85 | 28.34 |
| | | HT | −0.02 | 3.42 | −0.09 | 13.67 | 32.21 | 7.61 | 60.18 |
| MIC | | N | 0.012 | 0.160 | 0.049 | 0.641 | 55.64 | 8.31 | 36.05 |
| | | HT | 0.001 | 0.065 | 0.006 | 0.26 | 48.73 | 1.43 | 49.84 |
| SF | | N | 0.055 | 0.254 | 0.221 | 1.019 | 51.65 | 23.79 | 24.56 |
| | | HT | 0.001 | 0.28 | 0.006 | 1.104 | 28.19 | 17.42 | 54.39 |
| RD | | N | −0.070 | 8.742 | −0.282 | 34.969 | 24.01 | 15.34 | 60.65 |
| | | HT | 0.08 | 1.26 | 0.35 | 5.06 | 25.56 | 32.80 | 41.64 |
| UHML | | N | 0.028 | 3.487 | 0.113 | 13.951 | 45.48 | 0.28 | 54.24 |
| | | HT | 0.000 | 1.91 | 0.003 | 7.66 | 37.74 | 5.28 | 56.98 |
| UI | | N | −0.041 | 12.779 | −0.1664 | 51.117 | 29.19 | 12.31 | 58.50 |
| | | HT | 0.30 | 0.24 | 1.21 | 0.99 | 56.01 | 20.62 | 23.37 |
| H$_2$O$_2$ | | N | −0.143 | 7.564 | −0.571 | 30.25 | 14.66 | 18.48 | 66.86 |
| | | HT | 0.16 | 3.10 | 0.64 | 12.40 | 38.40 | 18.42 | 43.18 |
| CAT | | N | −4.909 | 790.741 | −19.638 | 3162.967 | 35.70 | 3.63 | 60.67 |
| | | HT | −8.99 | 789.23 | −35.94 | 3156.93 | 11.25 | 24.83 | 63.93 |
| POD | | N | 11.245 | 263.02 | 44.981 | 1052.10 | 47.86 | 11.18 | 40.96 |
| | | HT | −0.83 | 56.73 | −3.32 | 226.94 | 19.66 | 14.57 | 65.77 |
| SOD | | N | 1.386 | 31.899 | 5.547 | 127.59 | 29.73 | 27.80 | 42.47 |
| | | HT | 1.27 | 12.28 | 5.11 | 49.12 | 37.25 | 7.27 | 55.48 |
| TSP | | N | 0.033 | 0.777 | 0.132 | 3.108 | 47.66 | 5.09 | 47.25 |
| | | HT | −1.01 | 25.72 | −4.04 | 102.90 | 11.49 | 0.31 | 88.21 |
| Chlo.C | a | N | 0.000 | 0.056 | 0.003 | 0.224 | 18.73 | 29.78 | 51.50 |
| | b | N | −0.000 | 0.004 | −0.000 | 0.016 | 18.83 | 22.94 | 58.23 |
| | a | HT | −0.000 | 0.010 | −0.0007 | 0.041 | 19.64 | 14.96 | 65.4 |
| | b | HT | 0.00 | 0.0011 | 0.0002 | 0.0045 | 35.72 | 21.11 | 43.17 |
| Caro. | | N | −0.000 | 0.002 | −0.000 | 0.010 | 30.09 | 12.68 | 57.23 |
| | | HT | 0.000 | 0.003 | 0.000 | 0.01 | 20.51 | 29.23 | 50.26 |

PH, plant height (cm); TNB, number of bolls; BW, boll weight (g); SCY, seed cotton yield (g); GOT, ginning outturn (%); H$_2$O$_2$, hydrogen peroxide (µmol/g); CAT, catalase (U mg$^{-1}$ protein); POD, peroxidase (U mg$^{-1}$ protein); SOD, super-oxidase dismutase (U mg$^{-1}$ protein); TSP, total soluble protein (mg g$^{-1}$ FW); Chl a & b, chlorophyll contents a & b (mg g$^{-1}$ FW); Caro., carotenoids (mg g$^{-1}$ FW); SF, short fiber contents (%); STR, fiber strength (g/tex); MIC, MIC value (unit); RD, reflectance; UI, uniformity index (%); UHML, upper half mean length (mm).

**Table 10.** First year heterosis effects of various traits under normal and high temperature stress conditions.

| Crosses | Trt | PH | TNB | BW | SCY | GOT% | $H_2O_2$ | CAT | POD | SOD |
|---|---|---|---|---|---|---|---|---|---|---|
| Ghuari-1 × Fb-Falcon | N | −10.98 * | −31.16 ** | −11.29 ns | −29.07 ** | 24.86 ** | −0.31 ns | −37.09 ** | −57.18 ** | 31.62 * |
| | HT | −0.65 ns | −10.29 ns | 12.07 * | −23.17 * | −7.43 ns | −20.17 ns | −12.36 ** | −0.00 ns | 3.36 ns |
| Ghuari-1 × Fb-Smart 1 | N | −9.27 * | −12.02 ns | −3.23 ns | −10.83 ns | 15.01 ** | −41.81 * | −6.24 ns | −23.92 ** | −2.33 ns |
| | HT | −6.37 ns | −21.22 ns | −0.00 ns | −26.36 ns | 14.44 ns | 3.36 ns | −14.38 ** | −2.03 ns | −3.84 ns |
| Ghuari-1 × Jsq White Gold | N | 0.00 ns | −9.57 ns | 0.00 ns | −7.79 ns | 7.17 ns | −2.82 ns | −3.76 ns | −6.32 ns | 13.26 ns |
| | HT | 1.56 ns | 2.25 ns | −3.45 ns | −12.26 ns | −1.86 ns | 14.71 ns | −1.86 ns | −0.05 ns | −7.01 ns |
| Badar-1 × Fb-Falcon | N | −18.97 ** | −10.4 ns | 6.67 ns | −10.13 ns | −0.57 ns | 15.15 ns | −38.99 ns | −45.22 ** | −9.49 ns |
| | HT | −3.92 ns | 22.41 ns | 10.71 ns | 20.83 ns | −7.30 ns | −26.81 * | 8.06 * | 2.73 ns | −4.24 ns |
| Badar-1 × Fb-Smart 1 | N | −9.00 * | −14.06 ns | 1.67 ns | −16.00 ** | 5.56 ns | −31.82 ns | −58.65 ** | −54.47 ** | −8.76 ns |
| | HT | −3.46 ns | 18.75 ns | 8.93 ns | −8.91 ns | 8.68 ns | −33.62 * | −4.24 ns | −3.01 ns | −15.07 ** |
| Badar-1 × Jsq White Gold | N | −1.27 ns | 27.08 * | −1.67 ns | −21.40 ** | 0.67 ns | −46.97 ns | −45.51 * | −49.06 ** | 1.46 ns |
| | HT | −1.70 ns | 41.96 * | 3.57 ns | −2.44 ns | 2.17 ns | −31.06 * | −3.84 ns | −1.16 ns | −11.22 * |
| Eagle-2 × Fb-Falcon | N | −8.28 * | 6.68 ns | 7.02 ns | 4.83 ns | −6.74 ns | −36.27 * | 5.39 ns | 43.40 ** | 7.88 ns |
| | HT | −9.40 ** | 51.14 ** | −0.00 ns | 19.98 * | −1.67 ns | 8.87 ns | 15.93 ** | 1.50 ns | −4.74 ns |
| Eagle-2 × Fb-Smart 1 | N | −11.30 ** | 2.29 ns | 10.53 ns | −4.83 ns | 14.13 * | 12.75 ns | −64.29 ** | 126.57 ** | 12.90 ns |
| | HT | −19.54 ** | 21.21 ns | −10.17 ns | −13.69 ns | 2.17 ns | 63.55 ** | −12.35 ** | −12.91 ** | −11.25 * |
| Eagle-2 × Jsq White Gold | N | −20.30 ** | −37.02 ** | 10.53 ns | −2.15 ns | 17.28 * | −24.51 ns | −35.45 ** | −9.12 ns | 17.22 ns |
| | HT | −21.63 ** | 13.64 ns | 0.00 ns | −14.35 ns | 12.69 ns | −7.88 ns | −9.15 ** | −13.46 ** | −0.76 ns |
| Ccri-24 × Fb-Falcon | N | −11.29 * | −5.96 ns | −14.52 * | −14.75 * | 8.52 ns | −29.41 ns | −36.98 ** | −34.42 ** | 0.00 ns |
| | HT | −12.37 ** | −24.84 ns | 20.37 ** | −42.67 ** | 1.24 ns | −15.23 ns | −13.94 ** | −1.98 ns | −2.07 ns |
| Ccri-24 × Fb-Smart 1 | N | −10.43 * | 14.91 ns | −4.84 ns | −13.14 * | 19.45 ** | −41.18 ns | −33.33 ** | −24.49 * | 3.94 ns |
| | HT | −14.06 ** | 4.90 ns | 16.67 ** | −40.83 ** | 10.92 ns | −14.06 ns | −13.54 ** | −2.85 ns | 6.39 ns |
| Ccri-24 × Jsq White Gold | N | −7.37 ns | −3.21 ns | −11.29 ns | −15.55 * | 22.19 ** | 27.06 ns | −67.78 ** | −61.97 ** | −23.26 ns |
| | HT | −12.48 ** | −17.65 ns | 11.11 ns | −39.07 ** | 20.49 ** | −44.53 ** | −0.84 ns | −10.39 ** | 5.80 ns |
| Fb-Shaheen × Fb-Falcon | N | −8.70 * | −22.94 ns | 1.89 ns | −6.85 ns | 1.39 ns | −10.87 ns | −55.95 ** | 14.99 ns | −26.67 * |
| | HT | −12.79 ** | −8.48 ns | 3.39 ns | −20.15 * | −2.94 ns | −44.72 ** | 2.58 ns | 0.28 ns | 2.38 ns |
| Fb-Shaheen × Fb-Smart 1 | N | −8.94 * | 25.29 ns | 18.87 * | 22.48 ** | 9.25 ns | 0.00 ns | 63.43 ** | 88.04 ** | −13.50 ns |
| | HT | −6.87 ns | −25.45 * | 1.69 ns | −22.16 * | −7.05 ns | 34.17 * | 1.43 ns | −1.08 ns | 1.76 ns |
| Fb-Shaheen × Jsq White Gold | N | 7.06 ns | 20.29 ns | 39.62 ** | 44.86 ** | 3.58 ns | 202.17 ** | 83.47 ** | 177.03 ** | 12.50 ns |
| | HT | 6.63 ns | 11.82 ns | 18.64 ** | 38.28 ** | −7.73 ns | 76.88 ** | 38.91 ** | 16.83 ** | 22.08 ** |

*, significance (α = 0.05); **, highly significant (α = 0.01); ns, no significant; PH, plant height (cm); TNB, number of bolls; BW, boll weight (g); SCY, seed cotton yield (g); GOT, ginning outturn (%); $H_2O_2$, hydrogen peroxide (μmol/g); CAT, catalase (U mg$^{-1}$ protein); POD, peroxidase (U mg$^{-1}$ protein); SOD, super-oxidase dismutase (U mg$^{-1}$ protein).

**Table 11.** First year heterosis effects of various traits under normal and high temperature stress conditions.

| Crosses | Trt | TSP | Chloro. Con | | Caro. | MIC | RD | SF | STR | UHML | UI |
|---|---|---|---|---|---|---|---|---|---|---|---|
| | | | a | b | | | | | | | |
| Ghuari-1 × Fb-Falcon | N | −32.38 ** | −20.59 ** | −32.61 ** | −34.62 ** | 7.29 ns | 0.32 ns | −2.34 ns | −14.09 ** | 3.63 ns | −4.66 ** |
| | HT | −27.25 ** | −21.43 * | −14.58 ns | 8.33 ns | 15.38 * | −0.81 ns | −3.11 ns | 5.55 ns | 3.63 ns | 1.99 ns |
| Ghuari-1 × Fb-Smart 1 | N | 0.00 ns | −5.51 ns | −8.70 ns | −3.85 ns | 12.50 ns | −3.56 ns | −18.29 ** | −14.85 ** | 6.87 ns | −5.24 ** |
| | HT | −32.75 ** | −21.43 * | −8.33 ns | 8.33 ns | 21.98 ** | −1.01 ns | −2.48 ns | 20.55 ** | 6.87 ns | 1.93 ns |
| Ghuari-1 × Jsq White Gold | N | 0.95 ns | −4.41 ns | −5.43 ns | −1.92 ns | −12.50 ns | −8.27 * | −16.94 * | −18.99 ** | 4.01 ns | −3.89 * |
| | HT | −13.50 ns | −14.29 ns | 4.17 ns | 16.67 ns | −0.00 ns | −5.34 ** | 6.83 * | 8.32 ns | −1.72 ns | 3.20 * |
| Badar-1 × Fb-Falcon | N | −27.96 ** | −14.77 ns | −23.38 ** | −30.23 * | −1.80 ns | −3.22 ns | −4.09 ns | −12.77 ** | 10.42 * | −7.95 ** |
| | HT | 15.30 ns | 18.18 ns | 23.08 ns | 0.00 ns | −4.27 ns | −2.55 ns | 7.43 * | 4.48 ns | 2.40 ns | −3.86 ** |
| Badar-1 × Fb-Smart 1 | N | −37.98 ** | −4.22 ns | 19.48 * | 32.56 ** | −2.70 ns | −0.98 ns | −23.43 ** | −6.84 ns | 11.42 ** | −6.76 ** |
| | HT | −17.13 ns | −18.88 ns | −20.51 ns | −21.43 * | −5.98 ns | 1.62 ns | 18.92 ** | 1.49 ns | 9.42 * | 1.13 ns |
| Badar-1 × Jsq White Gold | N | −34.41 ** | −27.43 ** | −28.57 ** | −31.27 ** | −2.70 ns | 1.92 ns | −28.96 ** | −10.49 * | 11.02 * | −7.88 ** |
| | HT | −7.34 ns | 0.00 ns | 2.56 ns | −7.14 ns | −23.08 ** | −0.73 ns | 14.86 ** | 12.94 * | 3.01 ns | 0.65 ns |
| Eagle-2 × Fb-Falcon | N | 16.67 ns | 3.72 ns | 10.94 ns | 23.53 ns | 19.17 ns | −0.26 ns | −25.51 ** | −3.55 ns | 11.59 ** | −8.45 ** |
| | HT | 9.26 ns | 16.67 ns | 14.58 ns | 7.14 ns | −11.70 * | −1.99 ns | 5.70 ns | 4.17 ns | 4.27 ns | −1.98 ns |
| Eagle-2 × Fb-Smart 1 | N | 64.47 * | 56.74 ** | −17.19 ns | 26.32 * | 23.40 ** | −3.64 ns | −38.78 ** | 8.04 ns | 15.70 ** | −2.82 ns |
| | HT | −34.66 ** | −16.67 ns | −25.00 * | −28.57 ** | 3.19 ns | −1.13 ns | −3.80 ns | 12.74 * | −12.82 ** | −3.61 ** |
| Eagle-2 × Jsq White Gold | N | 0.00 ns | −12.56 ns | −4.69 ns | 0.00 ns | 14.74 * | −0.97 ns | −16.84 ** | −1.55 ns | 6.36 ns | −4.79 ** |
| | HT | −20.90 * | 25.0 * | −27.08 * | −14.29 ns | 17.71 ** | −5.57 ** | 17.72 ** | 15.82 * | 10.49 * | −1.40 ns |

**Table 11.** *Cont.*

| Crosses | Trt | TSP | Chloro. Con | | Caro. | MIC | RD | SF | STR | UHML | UI |
|---|---|---|---|---|---|---|---|---|---|---|---|
| | | | a | b | | | | | | | |
| Ccri-24 × Fb-Falcon | N | −20.79 * | −14.34 * | −18.29 * | −20.41 * | 29.21 ** | −7.41 * | 0.00 ns | 16.12 ** | −10.36 * | −4.00 * |
| | HT | −25.27 ** | −8.33 ns | −19.57 ns | 8.33 ns | 20.88 ** | −1.54 ns | 4.45 ns | 8.80 ns | −10.36 ns | 0.48 ns |
| Ccri-24 × Fb-Smart 1 | N | −16.83 ns | −12.75 ns | −13.41 ns | −16.33 ns | 42.05 ** | 3.28 ns | −13.48 * | 26.27 ** | −6.91 ns | −4.25 * |
| | HT | 23.12 * | 8.33 ns | −28.26 * | 8.33 ns | 30.00 ** | −5.54 ** | 1.30 ns | 10.47 ns | −6.91 ns | 2.92 * |
| Ccri-24 × Jsq White Gold | N | −50.50 ** | −37.45 ** | −42.46 ** | −55.10 ** | 18.95 * | 1.09 ns | −6.01 ns | 14.59 ** | 0.52 ns | −9.46 ** |
| | HT | −46.24 ** | −0.00 ns | −21.74 ns | 0.00 ns | 16.67 ** | −8.01 ** | 7.14 ns | 24.09 * | −21.93 ** | 3.52 ** |
| Fb-Shaheen × Fb-Falcon | N | −36.00 ** | −30.61 ** | −23.44 * | −27.03 ** | 30.77 ** | −7.84 * | 14.62 * | 30.23 ** | −3.87 ns | −2.80 ns |
| | HT | 41.30 ** | 44.44 ** | 24.32 ns | −0.00 ns | 15.96 ** | −5.44 ** | 1.32 ns | 8.41 ns | −5.53 ns | 2.82 * |
| Fb-Shaheen × Fb-Smart 1 | N | 38.16 ** | 30.30 ** | 31.5 ** | 34.21 * | 42.86 ** | −6.09 ns | 1.71 ns | 13.73 ** | −8.44 * | 0.95 ns |
| | HT | 8.79 ns | 11.11 ns | −2.70 ns | 8.33 ns | 21.28 ** | −0.07 ns | 14.57 ** | 25.40 ** | −10.02 * | 3.00 ** |
| Fb-Shaheen × Jsq White Gold | N | 72.00 ** | 41.84 * | 46.88 ** | 70.27 ** | 44.21 ** | −7.78 * | −10.93 ns | 14.41 ** | 1.05 ns | −0.11 ns |
| | HT | 119.67 * | 77.78 * | 48.65 ** | 45.45 ** | 21.87 ** | −7.01 ** | 1.99 ns | 5.18 ns | −7.60 * | 0.96 ns |

*, significance ($\alpha = 0.05$); **, highly significant ($\alpha = 0.01$); ns, no significant; TSP, total soluble protein (mg g$^{-1}$ FW); Chl a & b, chlorophyll contents a & b (mg g$^{-1}$ FW); Caro., carotenoids (mg g$^{-1}$ FW); SF, short fiber contents (%); STR, fiber strength (g/tex); MIC, MIC value (unit); RD, reflectance; UI, uniformity index (%); UHML, upper half mean length (mm).

**Table 12.** Second year heterosis effects of various traits under normal and high temperature stress conditions.

| Crosses | Trt | PH | TNB | BW | SCY | GOT% | H₂O₂ | CAT | POD | SOD |
|---|---|---|---|---|---|---|---|---|---|---|
| Ghuari-1 × Fb-Falcon | N | −12.54 * | −29.73 * | −11.29 * | −26.53 ** | 29.34 ** | −21.39 ns | −37.02 ** | −54.49 ** | 25.05 * |
| | HT | −4.55 ns | −15.19 ns | 4.92 ns | −21.11 * | −12.53 * | −19.35 ns | −12.54 ** | −0.00 ns | 3.95 ns |
| Ghuari-1 × Fb-Smart 1 | N | −9.27 * | −6.03 ns | −1.61 ns | −9.98 * | 18.08 ** | −35.84 ns | −7.36 * | −18.68 ** | 4.88 ns |
| | HT | −6.37 ns | −25.98 * | −3.28 ns | −28.42 ** | 12.95 * | −0.81 ns | −12.54 ** | 1.32 ns | −0.49 ns |
| Ghuari-1 × Jsq White Gold | N | 1.39 ns | −11.85 ns | 3.23 ns | −6.96 ns | 9.90 ns | −1.73 ns | −2.88 ns | −6.41 ns | 16.00 ns |
| | HT | −1.04 ns | −6.95 ns | −8.20 ns | −11.23 ns | 0.33 ns | 18.15 ns | −0.02 ns | −2.49 ns | 2.17 ns |
| Badar-1 × Fb-Falcon | N | −18.05 ** | −12.38 ns | 3.33 ns | −10.98 * | −2.49 ns | 13.64 ns | −39.23 ** | −41.24 ** | −17.04 ** |
| | HT | 3.23 ns | 12.30 ns | 5.08 ns | 21.95 * | −3.39 ns | −13.47 ns | 10.19 ** | 6.43 * | 0.15 ns |
| Badar-1 × Fb-Smart 1 | N | −9.00 ** | −15.84 ns | 5.00 ns | −16.79 ** | 1.72 ns | −31.82 ns | −53.52 ** | −55.40 ** | −8.15 ns |
| | HT | −0.60 ns | 13.11 ns | 3.39 ns | −1.70 ns | 8.76 ns | −24.08 ns | −2.54 ns | −1.62 ns | 1.06 ns |
| Badar-1 × Jsq White Gold | N | −0.66 ns | 15.84 ns | 1.67 ns | −23.08 ** | −2.20 ns | −44.70 ns | −45.60 ** | −47.50 ** | 0.00 ns |
| | HT | −0.40 ns | 38.52 ns | −1.69 ns | 8.92 ns | 3.19 ns | −13.47 ns | −2.99 ns | −1.62 ns | −4.52 ns |
| Eagle-2 × Fb-Falcon | N | −7.72 * | 4.86 ns | 6.56 ns | 5.78 ns | −3.48 ns | −28.82 ns | 2.16 ns | 31.40 * | 2.77 ns |
| | HT | −11.38 ** | 38.32 ** | −1.69 ns | 12.15 ns | −0.62 ns | −19.05 ns | 14.88 * | 10.98 ** | −1.75 ns |
| Eagle-2 × Fb-Smart 1 | N | −9.68 ** | 10.12 ns | 1.64 ns | −3.68 ns | 11.95 * | 24.12 ns | −58.40 ** | 104.42 * | 9.40 ns |
| | HT | −20.37 ** | 20.44 ns | −3.39 ns | −17.47 * | −2.08 ns | 21.61 ns | −11.51 ** | −6.64 * | −0.15 ns |
| Eagle-2 × Jsq White Gold | N | −22.27 ** | −31.91 * | 0.00 ns | −2.10 ns | 19.46 ** | −11.18 ns | −35.00 ** | −4.27 ns | 25.35 ** |
| | HT | −20.37 ** | 16.79 ns | −10.17 ns | −22.09 ** | 5.93 ns | −13.19 ns | −8.10 ** | −5.37 * | −2.83 ns |
| Ccri-24 × Fb-Falcon | N | −11.16 ** | −8.07 ns | −8.06 ns | −10.97 * | 4.22 ns | 1.80 ns | −38.29 ** | −29.78 ** | 0.00 ns |
| | HT | −14.38 ** | −2.17 ns | 12.07 ns | −36.45 ** | 0.21 ns | −15.23 ns | −14.73 ** | 2.63 ns | −9.80 * |
| Ccri-24 × Fb-Smart 1 | N | −8.22 * | 14.57 ns | −4.84 ns | −12.03 * | 18.02 ** | −40.12 * | −34.72 ** | −20.79 ** | 3.98 ns |
| | HT | −14.94 ** | 23.55 ns | 1.72 ns | −36.36 ** | 12.98 * | −14.06 ns | −14.33 ** | −4.33 ns | −4.46 ns |
| Ccri-24 × Jsq White Gold | N | −8.43 * | −3.14 ns | 5.36 ns | −15.28 ** | 19.67 ** | 26.35 ns | −65.85 ** | −55.47 ns | −11.04 ns |
| | HT | −13.38 ** | 5.80 ns | −10.34 ns | −31.63 ** | 20.49 ** | −13.28 ns | −1.75 ns | −8.28 ** | −4.51 ns |
| Fb-Shaheen × Fb-Falcon | N | −7.49 * | −21.67 ns | 14.29 * | 7.97 ns | 0.39 ns | −9.78 ns | −57.36 ** | 14.53 ns | −19.30 ** |
| | HT | −13.73 ** | 11.43 ns | −7.14 ns | −2.66 ns | −3.91 ns | 15.58 ns | 2.34 ns | −4.83 ns | −0.72 ns |
| Fb-Shaheen × Fb-Smart 1 | N | −6.18 ns | 21.11 ns | 32.14 ** | 20.96 ** | 9.16 ns | 1.09 ns | 61.60 * | 94.59 ** | −7.19 ns |
| | HT | −9.05 ** | 9.29 ns | 7.14 ns | 0.13 ns | −7.05 ns | 18.67 ns | 1.87 ns | −3.39 ns | 13.65 ** |
| Fb-Shaheen × Jsq White Gold | N | 8.66 * | 38.61 ** | 4.98 ns | 42.87 ** | 2.56 ns | 194.57 ** | 76.80 ** | 166.92 ** | 19.30 ** |
| | HT | 5.94 ns | 42.50 ** | 28.57 ** | 77.16 * | −8.71 ns | 81.91 ** | 37.68 ** | 13.29 ** | 16.91 ** |

*, significance ($\alpha = 0.05$); **, highly significant ($\alpha = 0.01$); ns, no significant; PH, plant height (cm); TNB, number of bolls; BW, boll weight (g); SCY, seed cotton yield (g); GOT, ginning outturn (%); H₂O₂, hydrogen peroxide (µmol/g); CAT, catalase (U mg$^{-1}$ protein); POD, peroxidase (U mg$^{-1}$ protein); SOD, super-oxidase dismutase (U mg$^{-1}$ protein).

**Table 13.** Second year heterosis effects of various traits under normal and high temperature stress conditions.

| Crosses | Trt | TSP | Chloro. Con a | Chloro. Con b | Caro. | MIC | RD | SF | STR | UHML | UI |
|---|---|---|---|---|---|---|---|---|---|---|---|
| Ghuari-1 × Fb-Falcon | N | 28.37 ns | −14.72 * | −19.79 ** | −34.62 ** | 7.29 ns | 0.32 ns | −2.34 ns | 1.34 ns | 3.63 ns | −4.66 ** |
| | HT | −29.02 ** | −20.00 * | −6.25 ns | 8.33 ns | 11.83 * | −2.82 ns | −14.2 ** | −1.01 ns | 15.84 * | −1.50 ns |
| Ghuari-1 × Fb-Smart 1 | N | 73.68 ** | 1.13 ns | −13.54 * | −3.85 ns | 12.50 ns | −3.56 ns | −18.29 ** | −0.80 ns | 6.87 ns | −5.34 ** |
| | HT | −34.39 ** | −22.96 * | −0.00 ns | 8.33 ns | 18.48 ** | 5.54 ** | −6.21 ns | 21.25 ** | 16.74 * | 0.06 ns |
| Ghuari-1 × Jsq White Gold | N | 9.47 ns | 1.89 ns | −11.46 ns | −1.92 ns | −12.50 ns | −8.27 * | −16.94 * | 30.50 ** | 4.01 ns | −3.89 * |
| | HT | −15.61 ns | −8.15 ns | 8.33 ns | 16.67 ns | 8.60 ns | −1.01 ns | 2.48 ns | 8.60 ns | 13.50 * | 3.82 ns |
| Badar-1 × Fb-Falcon | N | 16.78 ns | −8.79 ns | −30.86 ** | −30.23 * | −1.80 ns | −3.22 * | −4.09 * | 17.52 ns | 10.42 * | −7.95 ** |
| | HT | 0.92 ns | 17.82 ns | 33.33 * | 0.00 ns | 18.56 ** | −6.01 ** | 8.84 * | −2.23 ns | −9.07 ns | −2.67 ns |
| Badar-1 × Fb-Smart 1 | N | −0.00 ns | 3.35 ns | 6.82 ns | 32.56 ** | −2.70 ns | −0.98 ns | −23.43 ** | 6.39 ns | 11.42 ** | −6.76 ** |
| | HT | 12.58 ns | −18.81 ns | −2.56 ns | −21.43 ** | 9.28 * | −2.55 ns | 18.75 ** | 1.54 ns | 5.10 ns | 0.53 ns |
| Badar-1 × Jsq White Gold | N | −2.41 ns | −24.69 ** | −37.50 ** | −37.21 ** | −2.70 ns | 1.29 ns | −28.96 ** | 7.48 ns | 11.02 * | −7.88 ** |
| | HT | −12.58 ns | −1.98 ns | 12.82 ns | −7.14 ns | −0.00 ns | −1.08 ns | 16.08 ** | 6.52 ns | 2.84 ns | 1.84 ns |
| Eagle-2 × Fb-Falcon | N | −8.70 ns | 8.00 ns | −8.43 ns | 23.53 ns | 19.15 * | −0.26 ns | −25.52 ** | −1.98 ns | 11.59 ** | −8.45 ** |
| | HT | 14.55 ns | −14.29 ns | 18.75 ns | 15.38 ns | 10.64 * | −2.74 ns | 8.44 * | −6.00 ns | −3.43 ns | 2.94 ns |
| Eagle-2 × Fb-Smart 1 | N | 35.78 ns | 53.89 ** | −12.50 ns | 26.32 * | 23.40 ** | −3.46 ns | −38.78 ** | −2.74 ns | 15.70 ** | −2.82 ns |
| | HT | −21.43 * | −26.79 * | −25.00 * | −15.38 ns | 8.51 ns | −0.07 ns | −1.30 ns | 2.92 ns | −6.86 ns | 6.55 ** |
| Eagle-2 × Jsq White Gold | N | 10.87 ns | −16.44 * | −14.46 * | 0.00 ns | 14.74 * | −0.97 ns | −16.84 | −1.52 ns | 6.36 ns | −4.79 ** |
| | HT | −36.77 ** | −25.95 * | −27.08 * | −7.69 ns | 20.21 ** | 0.81 ns | 20.78 ** | 3.57 ns | 8.38 ns | 3.42 ns |
| Ccri-24 × Fb-Falcon | N | 3.09 ns | −13.18 * | −18.29 * | −20.41 * | 29.21 ** | −7.41 * | 0.00 ns | 30.53 ** | −10.3 ** | −4.00 ** |
| | HT | −4.64 ns | −27.48 ** | −26.92 * | 0.00 ns | 20.58 ** | −1.54 ns | 6.62 ns | 5.77 ns | −0.00 ns | −0.70 ns |
| Ccri-24 × Fb-Smart 1 | N | −18.56 ns | −10.85 ns | −7.32 ns | −16.33 ns | 42.05 ** | 3.28 ns | −13.48 * | 49.21 ** | −6.91 ns | −4.25 ns |
| | HT | 54.97 ** | −29.77 * | −34.62 * | −7.69 ns | 27.17 ** | −5.54 ** | 1.99 ns | 2.27 ns | 10.22 ns | 4.04 ns |
| Ccri-24 × Jsq White Gold | N | −38.41 ns | −39.15 ** | −32.93 ** | −55.10 ** | 18.95 * | 1.09 ns | −6.01 ns | 39.53 ** | 0.52 ns | −9.46 ** |
| | HT | −17.22 ns | −29.77 * | −42.31 ** | −7.69 ns | 21.51 ** | −7.34 ** | 8.61 * | −0.87 ns | 0.61 ns | 2.28 ns |
| Fb-Shaheen × Fb-Falcon | N | −26.88 ns | −29.56 ** | −33.73 ** | −27.03 * | 30.77 ** | −7.84 * | 14.62 * | 21.82 * | −3.87 ns | −2.80 ns |
| | HT | 56.77 ** | 10.11 ns | 24.32 ns | 0.00 ns | −3.51 ns | −5.44 ** | 1.32 ns | 10.20 ns | 1.41 ns | 4.02 ns |
| Fb-Shaheen × Fb-Smart 1 | N | −19.35 ns | 29.06 ** | 1.20 ns | 34.21 * | 42.86 ** | −6.09 ns | 1.71 ns | 12.88 ns | −8.44 * | 0.95 ns |
| | HT | 14.41 ns | 12.94 ns | −2.70 ns | 0.00 ns | 1.75 ns | −0.07 ns | 14.57 ** | 1.15 ns | −1.41 ns | 7.10 ** |
| Fb-Shaheen × Jsq White Gold | N | 27.96 ns | 36.95 ** | 14.46 * | 70.27 ** | 44.21 ** | −7.78 * | −10.93 ns | 34.35 ** | 1.05 ns | −0.11 ns |
| | HT | 109.16 ** | 75.29 ** | 48.65 ** | 23.08 ** | 2.63 ns | −5.67 ** | 1.99 ns | −3.29 ns | −4.22 ns | 5.83 * |

\*, significance ($\alpha$ = 0.05); \*\*, highly significant ($\alpha$ = 0.01); ns, no significant; TSP, total soluble protein (mg g$^{-1}$ FW); Chl a & b, chlorophyll contents a & b (mg g$^{-1}$ FW); Caro., carotenoids (mg g$^{-1}$ FW); SF, short fiber contents (%); STR, fiber strength (g/tex); MIC, MIC value (unit); RD, reflectance; UI, uniformity index (%); UHML, upper half mean length (mm).

## 4. Discussion

The success of breeding program for crop improvement is largely dependent on the ability of genetic material to transfer the favorable traits into its descendants [22,23]. The genetic material used in breeding program, should undergo critical selection procedure to assess its inherent potential. To study the genetics of heat tolerance, five heat tolerant and three sensitive cotton genotypes were crossed in lines × testers mating designs and 15 hybrids were developed. It is pertinent to mention that it offers valuable knowledge regarding genetic architecture of inheritance pattern from its combining ability estimates [24]. The results of present research clearly demonstrate the remarkable impacts of high temperature stress on various morphological, physicochemical and fiber quality traits of all cotton genotypes. In a stress breeding program, chlorophyll contents measurement plays a vital role for identifying the tolerant genotypes [3]. This suggests that heat tolerant genotypes correlate with the higher contents of the chlorophyll [25]. Under heat stress, the reduced chlorophyll contents can lead to increase in production of $H_2O_2$. The strong negative association between $H_2O_2$ and chlorophyll contents causes oxidative stress under heat treatment, which suggests that decrease in chlorophyll contents leads to the production of less photosynthates which in turn reduces the seed cotton yield [13]. Under heat treatment, abundant reactive oxygen species are produced, which restricts the process of photosynthesis and ultimately, speed up the oxygen-induced cellular damage. Furthermore, plants lose their cellular homeostasis as a result of reactive oxygen species production under heat stress. The upregulation of antioxidants defensive enzymes such as peroxidase (POD) and

catalase (CAT) triggers the plants to respond to metabolize ROS [26]. Peroxidase (POD) and Catalase (CAT) detoxifies and convert $H_2O_2$ into $H_2O$ and $O_2$ in the cytosol and chloroplast of the plant cell, respectively. The proposed study strengthens the previous findings that the rise in temperature increases $H_2O_2$ production [27]. However, its negative effects were prevented due to the scavenging activity of catalase. The CAT and POD activities were reported to be higher under heat stress than control. It has also been observed a higher level of CAT activity in cotton leaves under heat stress conditions [28]. The genotypes containing higher level of POD and CAT were found to be optimum for $H_2O_2$ and declared as heat-tolerant [29]. The phenotypes of every organism vary in response to continually warming environment in order to maintain their endurance. Every genotype can make specific alterations in its genome to transcribe subsequent related phenotypes due to abrupt altering environmental conditions around the world. To sustain these fluctuations, it's a need of hour that researchers must search for such genotypes in existing germplasms of cotton. In context, the study was carried out to investigate heat-resistant genotypes from available assets. The modest diversity panel utilized in the study, with only eight cotton parental genotype, may preclude concluding in this regard.

The significant interaction between genotype × treatment for morphological, biochemical and fiber quality traits indicated the significant effects of high temperature on yield and fiber quality traits [30]. The heat stress × year × genotypes interaction showed non-significant interaction for plant height, number of bolls, boll weight, SCY, GOT%, CAT, POD, chlorophyll a, carotenoids, RD and UHML which indicates that these genotypes are stable over years under high temperature stress conditions [31]. The decrease in non-additive gene action value under heat stress revealed the larger role of SCA in heat tolerance [32]. Studies at genomic level should be conducted to assess the synergistic functions of specified genes responsible for heat tolerance for stacking in the advanced acclimatized genotypes to produce high yielding heat tolerant germplasm [33]. During both years, the variance due to SCA was higher and more significant than GCA for all traits reflecting the dominant role of non-additive type of gene action under both conditions [34]. The higher value for non-additive gene action showed that the line × tester had high SCA and few genes are involved in expression of the traits [35]. The combining ability analysis helps to assess average breeding value (GCA) of genetic material used as well as the genetic value due to the interaction between these specific genes in a cross combination (SCA) [36]. The SCA is due to non-additive gene effects whilst GCA is due to the additive gene's effects [34]. The abiotic stresses are usually pleiotropic, the biomass allocation and reproductions output (yield) in the face of drought, which is a stress typically correlated with heat in the face of current changing climatic conditions [37,38]. Different studies suggested that the combination of both stresses results in higher yield reduction than individual stress due to the decrease in photosynthetic rate, alteration in water and energy balance, and disturbance in sucrose metabolism and carbohydrates concentrations. Iqbal et al. reported that heat stress combined with drought also results in significant reduction in boll retention, boll weight and seed cotton yield [39]. Single stress can play either a predominant role, based on plant responses to it and the combined stress to be similar or even a protective role [40].

The FB-SHAHEEN was good combiner for plant height, boll weight, seed cotton yield, CAT, POD, SOD, TSP, chlorophyll contents, MIC, UI and UHML under both conditions. The Eagle-2 was good combiner for BW, SCY and SF under heat stress conditions. The CCRI-24 revealed higher GCA effects for GOT% and fiber strength. Among testers JSQ WHITE GOLD had good GCA estimates for plant height, SCY, CAT, POD and SF under both conditions. The results demonstrate that the lines with higher GCA estimates for particular traits can be used in breeding program for the development of heat tolerant genotypes by hybridization followed by selection breeding [41]. The SCA results revealed that the cross combination of FB-Shaheen × JSQ White Gold exhibited significant and positive SCA estimates for plant height, NB, BW, SCY, CAT, POD, SOD, chlorophyll contents, carotenoids and MIC under both conditions. The cross combination of Eagle-2 × JSQ

White Gold revealed significant positive SCA estimates for short fiber index, UHML and uniformity index under heat stress conditions. Under heat stress conditions, the cross combination of CCRI-24 × Fb-Falcon, Ghuari-1 × FB-Smart1 and FB-Shaheen × FB-Falcon revealed positive and significant SCA effects for reflectance, MIC and fiber strength, respectively. Some of the cross combinations which exhibited higher positive SCA effects, it is not necessary that parents have good GCA [42]. Different studies indicates that good × poor, poor × poor and good × good parents resulted in hybrids that exhibits poor performance for the desired traits [43]. These findings indicated that those hybrids provide a good source to develop cotton germplasm for heat tolerance. Moreover, in our study hybrids with negative values under normal and heat stress suggested the existence of different genes with minor effects in each line or preponderance of epistasis [34]. Heterosis is the superiority in the performance of F1 hybrids over better parent [26]. In hybrids development program, heterosis is helpful to recognize superior parental combination. For most of the studied traits positive heterosis is desirable whilst for traits such as plant height and micronaire value, negative heterosis is desirable (Sing et al., 2012). The cross combinations of Eagle-2 × JSQ White Gold, Eagle-2 × Fb-Smart1, FB-Shaheen × FB-Falcon and CCRI-24 × Fb-Smart1 had significant and negative heterosis effects for plant height under high temperature stress conditions. The cross combination of Badar-1 × JSQ White Gold exhibited negative and significant heterosis estimates for MIC under heat stress conditions. During both years, the positive and significant heterosis effects for number of bolls, boll weight, SCY, CAT, POD, SOD, TSP, chlorophyll a & b, carotenoids and uniformity index were exhibited by Fb-Shaheen × JSQ White under both conditions [34,44]. For RD and UHML the cross combination of Ghuari-1 × Fb-Smart1 exhibited significant and positive heterosis estimates under high temperature stress conditions. For uniformity index and fiber strength Fb-Shaheen × Fb-Smart1 revealed significant and positive heterotic effects under heat stress conditions. The significant improvement in fiber quality traits by heterosis breeding were also reported by [34,45,46]. These crosses must be considered for hybrid development program to exploit better parent heterosis for the mentioned set of traits under heat stress conditions [47]. The presence of variation in the performance of the parents compared to hybrid development programs can be attributed to differences in the genetic constitution of the plants and their specific interaction with the prevailing environment. The presence of non-additive gene action for all studied characters revealed that the available genetic material can be a good option for the hybrid development [34]. Moreover, inter-specific crossing among wild relatives, landraces and elite cotton may offer an avenue to develop heat tolerant cotton genotypes [48,49]. Modern analytical approaches such as genomic prediction, machine learning, multi trait gene editing can accurately speed up the pre- and breeding endeavors to increase crop adaptability and yield in the face of rising temperature due to global warming. Future global fiber and food security involves the careful utilization of wild germplasm by using big data analytics developed through transdisciplinary approaches with open-source data and long-term funding [50]. Currently, leading cotton producing countries such as China, India, USA have attained higher production than Pakistan by developing cotton hybrids by using these approaches which are in perpetual evolutionary process [51]. The hybrid development of cotton in Pakistan is at the early stage in Pakistan and needs to be revitalized to enhance the cotton production in the country.

## 5. Future Perspective

The cotton hybrids identified in the present study can be utilized as a source of heat tolerance induction in high yielding germplasm acclimatized to the dynamic climate. Introgression of these traits can be accomplished by identifying the marker linked with the genes conferring heat tolerance in these genes through marker assisted parental selection and genomic assisted backcrossing can be performed for the effective transfer of those genes in the current high yielding germplasm. The transfer of the gene can be carried out by taking advantage of modern genome editing techniques such as CRISPR-Cas9 which does

not generate GMOs and is being widely accepted as non-transgenic [52]. However, there are a few obstacles in its applicability which are required to be removed (i.e., recalcitrant tissue culture in cotton after the CRISPR-Cas9 application has yet to be established and there is need to minimize the yield trade off after the induction of abiotic stress tolerance in the cotton germplasm). Future cotton hybrid breeding tailored to changing climatic conditions should be transdisciplinary and take the effective use of predictive breeding, genome editing coupled with identified heat tolerant germplasm [48].

## 6. Conclusions

High temperature stress badly affects the cotton production in Pakistan. All of the studied traits in this research were governed by non-additive gene action. The parents, which exhibited significant GCA effects in desirable direction for different traits under high temperature stress condition could be exploited for development of new synthetic varieties. Based on information from biometrical approaches used herein, FB-Shaheen × JSQ White Gold was best for most of the yield and fiber quality traits under normal and heat stress conditions. Potential genotypes can be efficiently employed in future cotton breeding programs to improve cotton crop yield and productivity by enhancing their heat tolerance to withstand the changing climate.

**Supplementary Materials:** The following supporting information can be downloaded at: https://www.mdpi.com/article/10.3390/agronomy12061310/s1. Table S1a: Temperature recorded in the tunnel during experiment, Table S1b: Weather data during crop season during 2018 and 2019; Table S2a: 1st year General combining ability effects under normal and high temperature stress conditions, Table S2b: 1st year Specific combining ability effects of crosses under normal and high temperature stress conditions; Table S3a: 1st year General combining ability effects under normal and high temperature stress conditions, Table S3b: 1st year: Specific combining ability effects of crosses under normal and high temperature stress conditions; Table S4a: 2nd year General combining ability effects under normal and high temperature stress conditions, Table S4b: 2nd year Specific combining ability effects of crosses under normal and high temperature stress conditions; Table S5a: 2nd year General combining ability effects under normal and high temperature stress conditions, Table S5b: 2nd year Specific combining ability effects of crosses under normal and high temperature stress conditions..

**Author Contributions:** M.M.Z., Y.Z. and M.A.F. wrote the initial draft of the manuscript and are equal contributors. A.A., H.F., M.H. and S.F. made all necessary corrections and carried out final editing of manuscript. A.S., A.R. and M.R. proof read the manuscript. Final approval for publication was given by the group leaders A.S., A.R. and M.R. All authors have read and agreed to the published version of the manuscript.

**Funding:** This work was supported by the Genetically Modified Organisms Breeding Major Project of China (2019ZX08010004–004) and the National Natural Science Foundation of China (31901579).

**Acknowledgments:** I am grateful to FB Genetics Four Brothers Group on providing us the expert opinion and facilities performing the analysis and writing the entire manuscript.

**Conflicts of Interest:** The authors confirm that there is no actual or potential conflicts of interest and also that the work described in this manuscript has not been published previously, that it is not under consideration for publication elsewhere. Its publication has been approved by all authors and implicitly by the responsible authorities where the work was carried out, and all persons entitled to authorship have been so named. The authors also confirm that, if accepted, it will not be published elsewhere including electronically in the same form, in English or in any other language, without the written consent of the copyright-holder.

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
