# Peer review of "Biochemical and Associated Agronomic Traits in Gossypium hirsutum L. under High Temperature Stress"

_agronomy, doi:10.3390/agronomy12061310_

Round 1
Reviewer 1 Report
Dear personalities:
As far as I am concerned, the article is original to be presented in the AGRONOMY, the document was written in form concise and the contributions are excellent to enlarge the knowledge in this sense. However, is important consider some observations realized.

Author Response
Response submitted

Reviewer 2 Report
The manuscript titled “Investigation of biochemical and associated traits in cotton under high temperature stress” provides the overview of different variance components which could contribute to high temperature tolerance in cotton. Author’s concluded SCA is more important than GCA and revealed these traits are under dominant gene action. However, manuscript lacks information in the methodology and discussion section. I would recommend authors to work on method section, providing more overview of statistical analysis, provide model equation and enlist all the source packages used for the analysis. There are many typos in the manuscript, please work on those. Authors can use this manuscript which was published in Agronomy as reference for working on the statistical analysis and equations (https://www.mdpi.com/2073-4395/11/12/2528/htm)
Author Response
Q1. The manuscript titled “Investigation of biochemical and associated traits in cotton under high temperature stress” provides the overview of different variance components which could contribute to high temperature tolerance in cotton. Author’s concluded SCA is more important than GCA and revealed these traits are under dominant gene action. However, manuscript lacks information in the methodology and discussion section. I would recommend authors to work on method section, providing more overview of statistical analysis, provide model equation and enlist all the source packages used for the analysis. There are many typos in the manuscript, please work on those. Authors can use this manuscript which was published in Agronomy as reference for working on the statistical analysis and equations.
Response: Endorsed, more details regarding methodology have been added for clear understanding and reproducibility.
Reviewer 3 Report
The work by Zafar et al. advances our understating of heat tolerance in cotton. It explores the typical, yet understudied, biomass allocation and reproductive output under heat stress using hybrid breeding of eight genotypes and their F1 hybrids. The report is well written and condensed, as well as technically appropriate. However, before being able to recommend acceptance, I invite authors to address the following amendments.
First, the introduction section properly closes with an explicit goal (L72). I just would recommend adding a short sentence before to emphasize the research gap that inspired pursuing this work. Also, close the section by explicitly referring in L72 to a research hypothesis (e.g. we hypothesize that hybrid breeding may leverage heat tolerance variation in diverse cotton genotypes). I recommend authors to explicitly using the terminology “hybrid breeding” throughout the text.
Second, concerning the results, tables are very well edited and insightful but some of them may be slightly overwhelming to readers given their size and information content. Therefore, I invite authors to depicting key results using barplots with error bars, and leave some of the full tables as supplemental material. This would improve its readability.
Third, the discussion, although perceptive, should embrace broader reflections on whether heat tolerance could be understood as a plastic or alternative a genetic adaptive strategy to cope with hot climates (L366 would be a good place for that). Please acknowledge that the modest diversity panel utilized in the study, with only eight cotton parental genotype, may preclude concluding in this regard.
Fourth, since abiotic stresses are usually pleiotropic, please also comment on biomass allocation and reproductions output (yield) in the face of drought, which is a stress typically correlated with heat in the face of changing climate (please refer to and include Front Genet 2019 10:954). What about envisioning combined drought + heat treatments to study the same cotton hybrid lines? (L387 would be a good place for that since Chiuta & Mutengwa 2020 refer to these combined heat drought treatments). Also related to this point on pleiotropy in abiotic stress responses, authors must briefly comment on the expected biomass/reproductive trade-off, evident in subtle ways such as plant architecture and seed nutrient accumulation. For instance, may cotton plant architecture in some of the hybrids help preventing soil desiccation and elevated night air temperatures? (L377 would be a good place to bring this point).
Fifth, although the paper provides good evidence into the agronomical and physiological mechanisms for heat tolerance in hybrid families of cotton, a major question that authors should prospect in their discussion is how cotton improvement for heat tolerance may unlock and effectively utilize hidden variation from the wild genepools (please refer to and include Genes 2021 12:556). Inter-specific crossing may offer an avenue in this regard that authors should acknowledge (please refer to and include Agronomy 2021 11:1978).
Last but not least, please envision any other recommendation by adding a short perspectives section before the conclusions (in L441). Specifically, what is the next step to study and utilize heat tolerance in F1 cotton hybrids? Perhaps recurrent backcrossing or advanced backcrosses? What is the future potential of marker assisted selection (MAS), and genomic-assisted back crossing (GABC) to leverage heat tolerance in diverse genotypes (just as discussed at Genes 2021 12:783 in the context of plant breeding for abiotic stress tolerance, a seminal review to be included).
Author Response
Response submitted

Round 2
Reviewer 2 Report
can accept this manuscript for the publication
Reviewer 3 Report
Great amendments. Authors have successfully considered as perspectives several recommendations to study in more detail, in the face of biochemical responses to heat stress, physiological correlations beyond agronomic traits. The work is suitable for MDPI's Agronomy